# Demonstration of SARS-CoV-2 Exposure in Korean Native Cattle and Korean Native Black Goats in Korea

**DOI:** 10.3390/ani13223498

**Published:** 2023-11-13

**Authors:** Da-Yun Bae, Ju-Hee Yang, Sung-Hyun Moon, Woo H. Kim, Dae-Sung Yoo, Choi-Kyu Park, Yeun-Kyung Shin, Hae-Eun Kang, Dongseob Tark, Yeonsu Oh, Ho-Seong Cho

**Affiliations:** 1Biosafety Research Institute, College of Veterinary Medicine, Jeonbuk National University, Iksan 54596, Republic of Korea; bdy7700@naver.com (D.-Y.B.); chunsu17@naver.com (S.-H.M.); 2Korea Zoonosis Research Institute, Jeonbuk National University, Iksan 54531, Republic of Korea; juheeppuing@jbnu.ac.kr (J.-H.Y.); tarkds@jbnu.ac.kr (D.T.); 3College of Veterinary Medicine & Institute of Animal Medicine, Gyeongsang National University, Jinju 52828, Republic of Korea; woohyun.kim@gnu.ac.kr; 4College of Veterinary Medicine, Chonnam National University, Gwangju 61186, Republic of Korea; shanuar@chonnam.ac.kr; 5College of Veterinary Medicine, Kyungbuk National University, Daegu 41566, Republic of Korea; parkck@knu.ac.kr; 6Foreign Animal Disease Division, Animal and Plant Quarantine Agency, Gimcheon 39660, Republic of Korea; shinyk2009@korea.kr (Y.-K.S.); kanghe@korea.kr (H.-E.K.); 7College of Veterinary Medicine & Institute of Veterinary Science, Kangwon National University, Chuncheon 24341, Republic of Korea

**Keywords:** COVID-19, SARS-CoV-2, Korean native cattle, Korean native black goat, reverse zoonosis

## Abstract

**Simple Summary:**

With the global spread of COVID-19, surveillance of companion animals living in close proximity to humans has been conducted due to the zoonotic nature of the causative agent, SARS-CoV-2. However, further research is still needed when it comes to livestock animals, particularly cattle and goats that are raised in grazing management systems as part of a group. In this study, nasal swabs and blood samples were collected from randomly selected Korean native cattle and Korean native black goats across nine different provinces in Korea. Of the 1798 animals tested, we found 1 cattle and 1 goat with detectable SARS-CoV-2 RNA, 54 Korean native cattle (4.60%) and 16 Korean native black goats (2.56%) with antibodies, and 51 Korean native cattle (4.34%) and 14 goats (2.24%) with virus-neutralizing antibodies. Infections have been found in various animals after initiating in wildlife and spreading widely among humans, and our results indicate the presence of antigens and antibodies in Korean native cattle and Korean native black goats. This suggests a potential reverse zoonosis, where humans may be infected through a widespread epidemic. Therefore, continuous monitoring is crucial, and additional testing should be conducted on a broader range of animals.

**Abstract:**

The COVID-19 pandemic is caused by the zoonotic SARS-CoV-2 virus. A wide range of animals that interact with humans have been investigated to identify potential infections. As the extent of infection became more apparent, extensive animal monitoring became necessary to assess their susceptibility. This study analyzed nasal swabs and blood samples collected from randomly selected Korean native cattle and Korean native black goats. The tests conducted included real-time qPCR to detect SARS-CoV-2 antigens, an ELISA to detect antibodies, and a plaque reduction neutralization test (PRNT) to determine the presence of neutralizing antibodies. Among the 1798 animals tested (consisting of 1174 Korean native cattle and 624 Korean native black goats), SARS-CoV-2 viral RNA was detected in one Korean native cattle and one Korean native black goat. ELISA testing revealed positive results for antibodies in 54 Korean native cattle (4.60%) and 16 Korean native black goats (2.56%), while PRNTs yielded positive results in 51 Korean native cattle (4.34%) and 14 Korean native black goats (2.24%). The presence of SARS-CoV-2 antigens and/or antibodies was identified in animals on farms where farmworkers were already infected. It is challenging to completely rule out the possibility of reverse zoonotic transmission from humans to livestock in Korea, although the transmission is not to the same extent as it is in highly susceptible animal species like minks, cats, and dogs. This is due to the limited geographical area and the dense, intensive farming practices implemented in these regions. In conclusion, continuous viral circulation between humans and animals is inevitable, necessitating ongoing animal monitoring to ensure public health and safety.

## 1. Introduction

Coronavirus disease 2019 (COVID-19), resulting from SARS-CoV-2 infection, is a human affliction that presumably originated from wildlife and escalated to a pandemic through extensive inter-human transmission. By 10 March 2023, global reports indicated that approximately 676 million people had contracted the virus, with the death toll surpassing 6.9 million [1]. This novel zoonotic pathogen has not only affected humans but also spread among animal populations. To date, there have been 775 cases documented across 36 countries since Hong Kong identified the initial instance of animal infection, including 29 species such as cats, dogs, minks, otters, pet ferrets, lions, tigers, pumas, snow leopards, gorillas, white-tailed deer, fishing cats, binturongs, South American coatis, spotted hyenas, Eurasian lynx, Canada lynx, hippopotamuses, hamster, mule deer, giant anteaters, West Indian manatees, black-tailed marmosets, common squirrel monkeys, mandrills, red foxes, big hairy armadillos, black-headed spider monkeys, and common woolly monkeys. Animals infected with the virus typically exhibit minimal clinical signs or pathological alterations. The majority of animal infections have been traced to initial human transmission suggesting instances of reverse zoonosis [2,3]. Emerging accounts of the virus passing from animals back to humans highlight this risk. For example, a Thai veterinarian contracted COVID-19 during the collection of a swab from a cat that belonged to a patient already diagnosed with the infection [4]. The absence of limited clinical signs may lead to underdiagnosed cases of SARS-CoV-2 infection in animal species, and it has been experimentally demonstrated that positive animals show low amounts of viral shedding over a relatively short period [5,6,7].

The susceptibility of domestic ruminants to SARS-CoV-2 is currently under investigation. These species are of particular interest since, as for other species, cattle and sheep angiotensin-converting enzyme 2 (ACE2) exhibits affinity to the receptor-binding domain (RBD) of the SARS-CoV-2 spike protein (S) [8,9]. In particular, livestock are often in close contact with farmworkers during grazing, with potential known SARS-CoV-2 wildlife reservoirs [10,11]. Korean native cattle, also known as Hanwoo, are a breed of cattle that is native to Korea [12]. They are highly valued for their meat quality which is famous for its marbling and tenderness. Korean native black goats, also known as Heugyeomso, are a breed of goats that is also indigenous to Korea. Korean native black goat meat is known for its unique flavor and tenderness. Both Korean native cattle and Korean native black goats have cultural and economic significance in Korea. Efforts are being made to preserve and promote these indigenous breeds to maintain their genetic diversity and support sustainable livestock farming practices. Additionally, research on their infection status and health management is essential to ensure the well-being of these animals and the safety of their products in light of various diseases, including SARS-CoV-2.

Grasping the role of domestic livestock, alongside the recognized significance of pets in SARS-CoV-2 transmission, is vital for safeguarding both animal and human health. Advancements in this area will contribute to pinpointing domestic reservoirs and forecasting the virus’s potential for endemic persistence [13]. Research indicates that dogs and cats can contract SARS-CoV-2; however, the susceptibility of farm animals to this particular virus and its prevalence in these populations under natural conditions remain underexplored [1].

The prevalence of infection among companion animals like dogs and cats has been extensively studied in various countries, including Italy, China, the United States, and Korea [13,14,15,16,17]. However, when it comes to the native animals of each country, specifically indigenous livestock, there is a lack of research on their infection status. This research is the inaugural effort to report the identification of SARS-CoV-2 antigens, antibodies, and virus-neutralizing antibodies in indigenous cattle and goats. Ongoing research is imperative to uncover potential new viral reservoirs and the emergence of new variants that could pose threats to human and animal health [11]. Further studies are crucial for enhancing our comprehension of the virus’s pathogenesis, zoonotic potential, and host-specific factors, and they are instrumental in advancing our capabilities to develop novel vaccines and targeted therapies [18]. This study aimed to determine the susceptibility of Korean native cattle and Korean native black goats to SARS-CoV-2 infection at farms nationwide during a COVID-19 outbreak among farmworkers.

## 2. Materials and Methods

### 2.1. Sample Collection Methodology

For this study, a representative sample of Korean native cattle and goats from nine provinces was selected. Recorded animal demographic information encompassed breed, sex, age, recent health history, respiratory symptoms (such as coughing, sneezing, nasal or ocular discharge, or conjunctivitis), and the COVID-19 status of the owners. A total of 1798 animals (comprising 1174 Korean native cattle and 624 goats) underwent sampling across Korea from September 2022 to August 2023. Nasal swabs were secured in a universal transport medium (UTM) provided by GDL Korea in Seoul, Korea. Concurrently, blood samples were allowed to clot and were then portioned into serum aliquots. These specimens were promptly transferred to our research facility on ice and preserved at −20 °C for subsequent analysis.

The collection of all specimens was meticulously performed by trained staff outfitted in personal protective gear, which included head covers, goggles, N95 masks, gloves, and disposable protective suits.

### 2.2. Virus and Cells

SARS-CoV-2 (IVCAS 6.7512) was isolated from a COVID-19 patient as previously described [19]. Vero E6 was purchased from the American Type Culture Collection (ATCC, Manassas, VA, USA) (ATCC^®^ CRL-1586™). All virus isolation experiments and neutralizing (VN) tests using SARS-CoV-2 were performed under biosafety level 3 (BSL3) conditions.

### 2.3. Extraction of Nucleic Acids and Real-Time qPCR for Reverse Transcription

The protocol for the extraction and identification of SARS-CoV-2 RNA from the collected swab samples proceeded as follows. UTMs were thoroughly mixed to produce 200 μL samples for the extraction process. Using a 16TU-CV19 Viral DNA/RNA Prep Kit from MiCo BioMed, based in Seoul, Korea, and their Veri-Q PREP M16 device, RNA was isolated as per the supplier’s protocol. Subsequently, a real-time qPCR assay for reverse transcription was employed to identify the ORF3a and nucleocapsid (N) genes of the SARS-CoV-2. This was performed using the nCoV-QM PCR kit, also from MiCo BioMed, alongside the Veri-Q PCR 316 QD-P100 system. The PCR mixture was composed of a 10 μL solution, which included 3 μL of a master mix consisting of polymerase, reverse transcriptase, buffer, and stabilizer, 1 μL of the primer/probe set, 1 μL of an internal positive control, and 5 μL of the extracted RNA, setting up the reaction. The thermal cycling parameters were set to reverse transcription at 50 °C for 10 min, initial denaturation at 95 °C for 3 min, followed by 45 cycles of denaturation at 95 °C for 9 s and annealing/extending at 58 °C for 30 s. A cycle threshold (Ct) value below 40 was indicative of a positive result for the detection of each targeted gene [13].

### 2.4. Enzyme-Linked Immunosorbent Assay (ELISA) for Antibody Detection

The serum samples were analyzed for antibodies (Abs) against the nucleocapsid protein (N protein) of SARS-CoV-2 using an indirect ELISA kit (ID Screen^®^ SARS-CoV-2 Double Antigen Multi-species, IDvet, Grabels, France) specifically developed for this purpose. This assay utilizes microwell plates that have been pre-coated with the recombinant N protein antigen. During the test, serum samples and a horseradish peroxidase (HRP)-conjugated N protein-recombinant antigen are introduced to the wells. The detection of SARS-CoV-2 Abs in the sera was based on the optical density (OD) readings at a wavelength of 450 nm. The assay was validated when the optical density of the positive control (OD_PC_) was ≥0.35 and the mean of the positive control (OD_PC_) to negative control (OD_NC_) ratio was greater than three. The optical density of each sample (OD_N_) was used to calculate the S/P ratio value (expressed as %), where S/P = 100 × (OD_N_ − OD_NC_)/(OD_PC_ − OD_NC_). ELISA-tested samples were considered positive if the S/P ratio was greater than 60%; doubtful when the P/S ratio ranged between 50 and 60%; and negative when the S/P score was lower than 50% [20].

ELISA for the differential diagnosis of Bovine coronavirus (BCoV) antibodies.

For BCoV antibody ELISA, a commercial indirect ELISA (SVANOVIR^®^ BCV-Ab, Svanova, Uppsala, Sweden) was used. The optical density (OD) reading at 450 nm was corrected via subtraction of the OD of the negative control antigen, and the percent positivity (PP value) was calculated as (corrected OD/positive control corrected OD) × 100. According to the test manuals, the recommended cut-off values of positive samples (>10 PP) were used as a starting point for both tests [21].

### 2.5. Plaque Reduction Neutralization Test (PRNT) for Assessing Neutralizing Antibodies

The plaque reduction neutralization test (PRNT) is a recognized as the definitive method for quantifying virus-neutralizing (VN) antibodies against SARS-CoV-2, standing out amongst various VN titration techniques [22,23]. This assay was conducted in a biosafety level 3 laboratory at the Korean Zoonosis Research Institute, Jeonbuk National University, Korea. Initially, 4 × 10^5^ Vero E6 cells per ml were plated into 12-well plates and cultured for 20 h at 37 °C with 5% CO_2_. Two days later, the virus was thawed and subjected to serial 10-fold dilutions ranging from 10^−1^ to 10^−5^. Concurrently, a 2% low-melting-point agarose solution was prepared in ultra-pure water, blended with 2× minimal essential medium (MEM; Gibco, Life Technologies Corporation, Carlsbad, CA, USA) supplemented with 4% fetal bovine serum (FBS, Gibco), ay a 1:1 ratio after two PBS washes. The cell layers were then infected with 100 μL of the virus dilutions in replicates. Following an hour’s incubation at 37 °C with 5% CO_2_, with intermittent shaking every 10 min, the infective medium was discarded, and cells were washed with PBS. An overlay mixture comprising 1% low-melting-point agarose, 1× MEM, and 2% FBS was then added and allowed to settle at room temperature for 10 min before incubation at 37 °C with 5% CO_2_ continued until a cytopathic effect (CPE) became apparent. Upon observing CPE, 1 mL of 10% formalin was introduced into the overlay to fix the cells for 1.5 h. After removing the staining solution, the number of viral plaques was determined and used to calculate the viral titer using the equation: Plaque-forming units per mL (Pfu/mL) = number of plaques/(dilution factor × volume of infection in mL). The serum neutralization titer is the reciprocal of the highest dilution resulting in an infection reduction of >50% (PRNT_50_). A titer of ≥1:10 was considered to be positive [24].

### 2.6. Analytical Evaluation of Data

Descriptive statistics were compiled for the collected samples based on their geographical origin to evaluate data integrity and distribution normalcy. To compare the detection rates of antigens and antibodies in Korean native cattle and goats, Fisher’s exact test was employed [25]. The statistical procedures were carried out with the aid of IBM SPSS Statistics 24.0 (IBM, Armonk, NY, USA). A *p*-value of less than 0.05 was set as the threshold for statistical significance.

## 3. Results

### 3.1. Detection of SARS-CoV-2 Antigen in Nasal Samples of Cattle and Goats

Real-time qPCR amplification results of the presence of SARS-CoV-2 in samples were presented in Figure A1 and interpreted in Table A1. In the evaluation of 1798 specimens utilizing real-time qPCR for SARS-CoV-2 RNA, which included samples from 1174 Korean native cattle and 624 goats, it was discovered that two samples were positive (Table 1). Within the Jeollabuk-do and Jeollanam-do regions, one Korean native cattle (sample number 51, 1 out of 130) and one goat (sample number 59, 1 out of 211), respectively, tested positive for the viral RNA. The cattle sample showed a Ct value of 34.627 for ORF 3a and 33.921 for the N gene. Similarly, the goat sample had Ct values of 31.821 for ORF3a and 32.512 for the N gene (Table 2).

### 3.2. Detection of Antibodies to SARS-CoV-2 in Serum Samples of Korean Native Cattle and Goats

Out of 1798 samples, which included 1174 from Korean native cattle and 624 from goats, tests were carried out for SARS-CoV-2 antibodies with ELISA and PRNTs. The ELISA identified antibodies to SARS-CoV-2 in 70 samples spread across nine provinces, as detailed in Table 1 and Table 2. Of these, 54 cattle samples (representing 4.60%) and 16 goat samples (2.56%) were found to be antibody-positive (Figure 1 and Table 1). Furthermore, to verify the specificity of the antibodies, sera from cattle and goats that tested positive for SARS-CoV-2 were checked for cross-reactivity with inactivated BCoV through an ELISA. The results demonstrated high specificity of the ELISA, indicated by the absence of serological cross-reactivity between SARS-CoV-2 and BCoV.

To confirm the neutralizing capability of sera that tested positive by ELISA, a PRNT specific to SARS-CoV-2 was utilized. Upon conducting neutralizing antibody assessments on the ELISA antibody-positive samples, it was found that five instances did not generate any neutralizing antibodies. Out of the 70 serum samples that were positive, 65 exhibited neutralization activity, with titers ranging between 1:10 and 1:160 (Table 2). The absence of neutralization activity in the remaining five samples could be due to the non-presence of specific neutralizing epitopes.

## 4. Discussion

Like numerous other devastating global outbreaks, SARS-CoV-2, initially transmitted from animals to humans, is inflicting severe health and economic consequences across the globe [5]. Instances of SARS-CoV-2 have been intermittently identified in domestic dogs and cats, typically those residing in proximity to COVID-19-posotive individuals [13,26,27,28].

Without clearly knowing the susceptibility and duration of SARS-CoV-2 infections in animals, the role of livestock like cattle and goats in the mechanics of transmission is still not fully understood. Despite the potential susceptibility of cattle and small ruminants to SARS-CoV-2, linked to the binding affinity between ACE2 and the virus’s RBD, there are few reports about the infection status in ruminants like cattle and goats. Moreover, there is no information available regarding the SARS-CoV-2 situation in Korean native livestock. In similar research to this in Italy, of 24 collected serum samples from lactating cows, 11 exhibited antibodies against the SARS-CoV-2 nucleocapsid protein, 14 exhibited antibodies against the SARS-CoV-2 spike protein, and 13 exhibited neutralizing antibodies for SARS-CoV-2 [11]. In Italian goats, 6 out of 124 goats (4.83%) were antibody-positive in an ELISA [29]. In the titrations of serum samples using antibody ELISA, given that both SARS-CoV-2 and BCoV are members of the betacoronavirus genus and considering BCoV’s close resemblance to the human coronavirus (HCoV) OC43 (betacoronavirus 1) [30], samples that yielded positive results for SARS-CoV-2 were subsequently tested for BCoV as well.

Despite the absence of SARS-CoV-2 viral RNA in all evaluated companion animals (182 dogs and 313 cats), the VN against the virus were present in 0.8% of dogs and 1.7% of cats in Southern Italy [31]. Reports from the Japanese Ministry of Agriculture, Forestry, and Fisheries to the World Organization for Animal Health (WOAH) indicated that dogs belonging to individuals with COVID-19 did not exhibit clinical signs and tested positive only up to the fourth day after symptom onset, turning negative from the fifth day onward [32,33,34].

In our study, among the 1798 livestock examined, only 2 were SARS-CoV-2 viral RNA positive, 70 were positive for antibodies, while all of them had COVID-19-positive owners. It has been demonstrated by experimental and natural infections that positive animals exhibit low amounts of viral shedding over a very short time. Therefore, serological examinations revealing a previous viral infection are a useful tool to investigate SARS-CoV-2’s host range and prevalence in animal species [29]. While the trend was global, South Korea also saw a significant increase in COVID-19 patients, a phenomenon from which even farm owners were not spared. Consequently, all the farm families we studies were confirmed cases, and this leads to a reasonable suspicion that the Korean native cattle and black goats they were raising could have been infected by them.

As it stands, human beings are the primary vectors for the spread of SARS-CoV-2, transmitting the virus not only amongst themselves but also to various animal species. Present evidence suggests that the likelihood of companion animals significantly contributing to the transmission of SARS-CoV-2 to humans is minimal [4]. Nevertheless, there is a notable body of research and case reports concerning companion animals in regions experiencing high rates of human infections [26,27].

Considering the vast array of pathogens, particularly those found in wildlife, along with the continuous evolution of viruses, the appearance of new zoonotic diseases is a certainty. Moreover, the interplay of human, animal, and environmental factors is a catalyst for the emergence of these diseases, potentially leading to future pandemic [35,36].

As SARS-CoV-2 viral RNA, indicative of the virus antigen, or antibodies against the virus were detected in the investigated animals, although it was not planned, all farmworkers at the surveyed farms were infected with COVID-19. This can cause continuous viral circulation between humans and animals; an initially zoonotic disease may convert to reverse zoonotic [4,5,31]. This study represents the inaugural documentation of SARS-CoV-2 infections among bovine and caprine populations in Korea. Given the significant transmissibility of SARS-CoV-2, there is a suggestion for the implementation of ongoing surveillance within animal populations to improve predictions of infection patterns.

## 5. Conclusions

In summary, this investigation indicates that domesticated species such as Korean native cattle and Korean native black goats are susceptible to SARS-CoV-2, potentially contracting the virus from humans. While there are no international guidelines mandating routine testing of SARS-CoV-2 in animals, findings from this research underscore the necessity of expanding monitoring efforts to include livestock, in addition to companion animals, for SARS-CoV-2 antibodies. Further exploration is warranted to comprehend the potential role of livestock as carriers or vectors in the propagation of SARS-CoV-2. In anticipation of future pandemics, establishing a robust surveillance network for domestic animals that frequently interact with humans is crucial.

## Figures and Tables

**Figure 1 animals-13-03498-f001:**
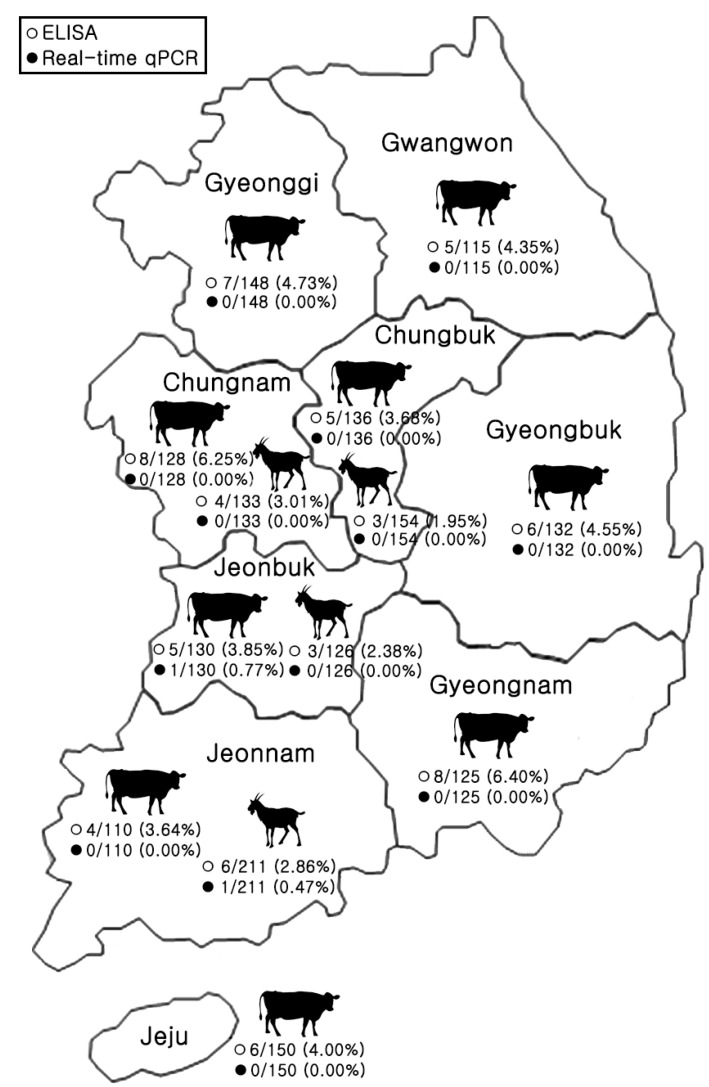
Geographical spread of Korean native cattle and goat samples analyzed by real-time qPCR and ELISA across Korea.

**Table 1 animals-13-03498-t001:** Prevalence of SARS-CoV-2 in Korean native cattle and goats by regions in Korea.

Region (Province)	qPCR		Total (%)	ELISA		Total (%)	PRNT		Total (%)
	Korean Native Cattle (%)	Korean Native Goat (%)		Korean Native Cattle (%)	Korean Native Goat (%)		Korean Native Cattle (%)	Korean Native Goat (%)	
Gyeonggi	0/148 (0.00)	-	0/148 (0.00)	7/148 (4.73)	-	7/148 (4.73)	7/148 (4.73)	-	7/148 (4.73)
Gangwon	0/115 (0.00)	-	0/115 (0.00)	5/115 (4.35)	-	5/115 (4.35)	5/115 (4.35)	-	5/115 (4.35)
Chungbuk	0/136 (0.00)	0/154 (0.00)	0/290 (0.00)	5/136 (3.68)	3/154 (1.95)	8/290 (2.76)	4/136 (2.94)	2/154 (1.30)	6/290 (2.07)
Chungnam	0/128 (0.00)	0/133 (0.00)	0/261 (0.00)	8/128 (6.25)	4/133 (3.01)	12/261 (4.60)	7/128 (5.47)	4/133 (3.01)	11/261 (4.22)
Gyeonbuk	0/132 (0.00)	-	0/132 (0.00)	6/132 (4.55)	-	6/132 (4.55)	6/132 (4.55)	-	6/132 (4.55)
Gyeongnam	0/125 (0.00)	-	0/125 (0.00)	8/125 (6.40)	-	8/125 (6.40)	7/125 (5.60)	-	7/125 (5.60)
Jeonbuk	1/130 (0.77)	0/126 (0.00)	1/256 (0.39)	5/130 (3.85)	3/126 (2.38)	8/256 (3.13)	5/130 (3.85)	3/126 (2.38)	8/256 (3.13)
Jeonnam	0/110 (0.00)	1/211 (0.47)	1/489 (0.20)	4/110 (3.64)	6/211 (2.86)	10/321 (3.12)	4/110 (3.64)	5/211 (2.37)	9/321 (2.80)
Jeju	0/150 (0.00)	-	0/150 (0.00)	6/150 (4.00)	-	6/150 (4.00)	6/150 (4.00)	-	6/150 (4.00)
Total	1/1174 (0.09)	1/624 (0.16)	2/1798 (0.11)	54/1174 (4.60)	16/624 (2.56)	70/1798 (3.89)	51/1174 (4.34)	14/624 (2.24)	65/1798 (3.62)

**Table 2 animals-13-03498-t002:** Detection of antigens and antibodies against SARS-CoV-2 in Korean native cattle and black goats by qPCR, ELISA and PRNT.

No.	qPCR (Ct)		ELISA		PRNT_50_	Background of Animal				
	ORF3a	N Gene	OD_450_	S/P Ratio Value (%)	Neutralization Titer	Species	Sex	Age (Year)	Province	COVID-19 Patient Owner
1	>40	>40	0.5216	73.59	1:20	Cattle	F	2	Gyeonggi	Yes
2	>40	>40	1.2381	184.48	1:40	Cattle	F	3	Gyeonggi	Yes
3	>40	>40	0.7526	109.34	1:40	Cattle	F	2	Gyeonggi	Yes
4	>40	>40	0.9178	134.91	1:20	Cattle	F	2	Gyeonggi	Yes
5	>40	>40	1.2501	186.34	1:40	Cattle	F	1	Gyeonggi	Yes
6	>40	>40	0.6274	89.96	1:20	Cattle	F	2	Gyeonggi	Yes
7	>40	>40	0.7391	107.25	1:40	Cattle	F	2	Gyeonggi	Yes
8	>40	>40	0.7147	103.47	1:20	Cattle	F	3	Gangwon	Yes
9	>40	>40	1.1398	169.27	1:40	Cattle	F	1	Gangwon	Yes
10	>40	>40	0.6793	98.00	1:20	Cattle	M	1	Gangwon	Yes
11	>40	>40	0.7418	107.67	1:40	Cattle	F	2	Gangwon	Yes
12	>40	>40	0.6731	97.04	1:20	Cattle	F	2	Gangwon	Yes
13	>40	>40	0.7419	107.68	1:20	Cattle	F	3	Chungbuk	Yes
14 *	>40	>40	0.6274	89.96	<1:10	Cattle	F	2	Chungbuk	Yes
15	>40	>40	0.6832	98.60	1:10	Cattle	M	1	Chungbuk	Yes
16	>40	>40	1.0361	153.22	1:20	Cattle	F	2	Chungbuk	Yes
17	>40	>40	0.9382	138.07	1:10	Cattle	F	4	Chungbuk	Yes
18	>40	>40	0.5787	82.43	1:20	Goat	M	1	Chungbuk	Yes
19 *	>40	>40	0.6579	94.68	<1:10	Goat	F	1	Chungbuk	Yes
20	>40	>40	0.6404	91.97	1:10	Goat	M	1	Chungbuk	Yes
21 *	>40	>40	0.6572	94.58	<1:10	Cattle	F	2	Chungnam	Yes
22	>40	>40	0.7944	115.81	1:40	Cattle	M	1	Chungnam	Yes
23	>40	>40	0.7340	106.46	1:20	Cattle	F	3	Chungnam	Yes
24	>40	>40	0.8872	130.17	1:40	Cattle	F	2	Chungnam	Yes
25	>40	>40	1.4052	210.35	1:80	Cattle	F	4	Chungnam	Yes
26	>40	>40	0.5977	85.37	1:10	Cattle	F	3	Chungnam	Yes
27	>40	>40	1.2712	189.61	1:80	Cattle	F	1	Chungnam	Yes
28	>40	>40	0.5193	73.23	1:20	Cattle	F	2	Chungnam	Yes
29	>40	>40	1.4263	213.61	1:80	Goat	M	1	Chungnam	Yes
30	>40	>40	0.6813	98.31	1:10	Goat	F	1	Chungnam	Yes
31	>40	>40	1.1251	166.99	1:40	Goat	M	1	Chungnam	Yes
32	>40	>40	0.8141	118.86	1:20	Goat	M	1	Chungnam	Yes
33	>40	>40	0.8222	120.11	1:40	Cattle	F	1	Gyeongbuk	Yes
34	>40	>40	0.6833	98.61	1:10	Cattle	M	1	Gyeongbuk	Yes
35	>40	>40	0.8614	126.18	1:20	Cattle	F	2	Gyeongbuk	Yes
36	>40	>40	1.0915	161.79	1:20	Cattle	M	1	Gyeongbuk	Yes
37	>40	>40	1.0813	160.22	1:40	Cattle	F	3	Gyeongbuk	Yes
38	>40	>40	1.0101	149.20	1:20	Cattle	F	2	Gyeongbuk	Yes
39	>40	>40	0.8381	122.57	1:20	Cattle	F	2	Gyeongnam	Yes
40	>40	>40	0.9112	133.89	1:40	Cattle	F	1	Gyeongnam	Yes
41 *	>40	>40	0.4657	64.94	<1:10	Cattle	F	1	Gyeongnam	Yes
42	>40	>40	0.9718	143.27	1:20	Cattle	F	1	Gyeongnam	Yes
43	>40	>40	1.0614	157.14	1:20	Cattle	M	1	Gyeongnam	Yes
44	>40	>40	1.1089	164.49	1:40	Cattle	F	1	Gyeongnam	Yes
45	>40	>40	1.0071	148.73	1:20	Cattle	F	1	Gyeongnam	Yes
46	>40	>40	0.8258	120.67	1:10	Cattle	F	1	Gyeongnam	Yes
47	>40	>40	0.8132	118.72	1:20	Cattle	M	1	Jeonbuk	Yes
48	>40	>40	1.5537	233.33	1:80	Cattle	M	1	Jeonbuk	Yes
49	>40	>40	0.7889	114.96	1:20	Cattle	F	1	Jeonbuk	Yes
50	>40	>40	1.5459	232.12	1:40	Cattle	F	1	Jeonbuk	Yes
51 **	>34.672	>33.921	1.1168	165.71	1:40	Cattle	F	1	Jeonbuk	Yes
52	>40	>40	1.2901	192.53	1:20	Goat	F	1	Jeonbuk	Yes
53	>40	>40	0.8071	117.78	1:20	Goat	M	1	Jeonbuk	Yes
54	>40	>40	0.4591	63.91	1:10	Goat	F	1	Jeonbuk	Yes
55	>40	>40	1.5612	234.49	1:160	Cattle	F	3	Jeonnam	Yes
56	>40	>40	0.8721	127.84	1:20	Cattle	F	1	Jeonnam	Yes
57	>40	>40	0.9712	143.17	1:20	Cattle	M	1	Jeonnam	Yes
58	>40	>40	0.7551	109.73	1:10	Cattle	F	2	Jeonnam	Yes
59 **	>31.821	>32.512	1.0213	150.93	1:20	Goat	M	2	Jeonnam	Yes
60	>40	>40	0.5941	84.81	1:10	Goat	M	1	Jeonnam	Yes
61	>40	>40	0.6201	88.83	1:20	Goat	F	1	Jeonnam	Yes
62 *	>40	>40	0.5404	76.50	<1:10	Goat	F	1	Jeonnam	Yes
63	>40	>40	0.8857	129.94	1:10	Goat	M	1	Jeonnam	Yes
64	>40	>40	0.8938	131.19	1:40	Goat	M	1	Jeonnam	Yes
65	>40	>40	1.2158	181.03	1:40	Cattle	M	1	Jeju	Yes
66	>40	>40	0.8062	117.64	1:10	Cattle	F	2	Jeju	Yes
67	>40	>40	0.8392	122.74	1:20	Cattle	F	1	Jeju	Yes
68	>40	>40	1.4531	217.76	1:80	Cattle	F	1	Jeju	Yes
69	>40	>40	0.7187	104.09	1:20	Cattle	F	1	Jeju	Yes
70	>40	>40	1.5215	228.35	1:40	Cattle	F	1	Jeju	Yes

* Cases where sera tested positive for antibodies via ELISA but did not exhibit neutralizing activity in the PRNT assay for SARS-CoV-2; ** cases where SARS-CoV-2 RNA was detected via qPCR.

## Data Availability

Data are contained within the article.

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
