# Peer review of "Demonstration of SARS-CoV-2 Exposure in Korean Native Cattle and Korean Native Black Goats in Korea"

_animals, 2023, doi:10.3390/ani13223498_

Round 1

Reviewer 1 Report

Comments and Suggestions for Authors

I express my gratitude to the authors for their interesting work and offer the authors a number of comments on the basis of which the article can be improved for understanding

1. In summary you are writng "we 23 found 1 cattle and 1 goat with detectable SARS-CoV-2 antigens", however, using the PCR method you determine not antigens, but the presence of viral RNA. From a biochemical point of view, an antigen is a molecule that specifically binds to an antibody. Antigens could be determined by the method sandwich ELISA.

The same in Discussion (line 268, 285)

2. When referring to the figure1A it would be better to indicate that it is located in the appendix (line 194). But it is not clear why the PCR test of a negative sample is given on it. There are several positive samples in the analyzes; it is more logical to present the amplification curves of these particular samples. Considering that you used a commercial kit, the provision of graphic information in general can be excluded. It is enough to indicate the kit name in the methods.

3. In the Table2 you need to indicate in the description of the table which samples you are considering in it. Intuitively, upon first viewing, this is not very clear and the reader has to look for explanations in the text.

4. Easy to understand graphical display of Korean native cattle and goats distribution is presented on Fig1, but it in the description of the figure it would be better to indicate the meaning of the points (ELISA and PCR) for a more unambiguous interpretation.

5. The first paragraph of the Discussion and the first two sentences of the second paragraph (lines 239-247) can be painlessly removed, all this has already been described in the Introduction

6. Line 268  - 1798 samples

7. Could you please explain why the methods indicate two ELISA methods (lines 141 and 155). In the results everywhere you simply refer to ELISA

8. It would be interesting to learn more about sample collection. The methods do not indicate how many farms in each province were surveyed or one from each. It is also interesting to know whether medical history and the presence of respiratory signs correlate with the obtained data.

Author Response

Response to reviewers’ comments

We are pleased to resubmit a revised manuscript (no. animals-2692205) entitled “Evidence of exposure to SARS-CoV-2 in Korean native cattle and Korean native black goats in Korea” is changed to “Evidence of exposure to SARS-CoV-2 in the native cattle and black goats in Korea” for reconsideration in Animals published by MDPI as an original manuscript. We have carefully evaluated the reviewer’s comments and have provided a point-by-point response below. Changes in the manuscript have been identified by colored font. We hope that the revised manuscript meets the reviewers’ expectations at Animals.

Reply to Academic Editor :

Reviewer #1.

Comments and Suggestions for Authors

I express my gratitude to the authors for their interesting work and offer the authors a number of comments on the basis of which the article can be improved for understanding.

  1. In summary you are writing "we 23 found 1 cattle and 1 goat with detectable SARS-CoV-2 antigens", however, using the PCR method you determine not antigens, but the presence of viral RNA. From a biochemical point of view, an antigen is a molecule that specifically binds to an antibody. Antigens could be determined by the method sandwich ELISA.

The same in Discussion (line 268, 285)

Thank you for your meticulous comments. What you pointed out has been changed in the main text and marked in blue.

  1. When referring to the figure1A it would be better to indicate that it is located in the appendix (line 194). But it is not clear why the PCR test of a negative sample is given on it. There are several positive samples in the analyzes; it is more logical to present the amplification curves of these particular samples. Considering that you used a commercial kit, the provision of graphic information in general can be excluded. It is enough to indicate the kit name in the methods.

Thank you for the reviewer’s meticulous comment and it was reflected in the main text in blue.

  1. In the Table2 you need to indicate in the description of the table which samples you are considering in it. Intuitively, upon first viewing, this is not very clear and the reader has to look for explanations in the text.

Thank you for the reviewer’s meticulous comment and it was reflected in the main text in blue.

  1. Easy to understand graphical display of Korean native cattle and goats distribution is presented on Fig1, but it in the description of the figure it would be better to indicate the meaning of the points (ELISA and PCR) for a more unambiguous interpretation.

We must mention cautiously, but the dots (white and black) used in that figure are already indicated as a legend in the top left corner of the figure. Please check to confirm.

  1. The first paragraph of the Discussion and the first two sentences of the second paragraph (lines 239-247) can be painlessly removed, all this has already been described in the Introduction

Thank you for the reviewer’s comment and it was reflected in the main text in blue.

  1. Line 268  - 1798 samples

Thank you for your meticulous comments. It has been changed in the main text and marked in blue.

  1. Could you please explain why the methods indicate two ELISA methods (lines 141 and 155). In the results everywhere you simply refer to ELISA

Thank you for your meticulous comments. BCoV antibody ELISA was used to confirm whether SARS CoV-2 antibodies and BCoV antibodies cross-react with each other. It has been changed in the main text and marked in blue.

  1. It would be interesting to learn more about sample collection. The methods do not indicate how many farms in each province were surveyed or one from each. It is also interesting to know whether medical history and the presence of respiratory signs correlate with the obtained data.

Thank you for your meticulous comments. For Korean native cattle, the number of farms sampled by province was Gyeonggi (30 farms), Gangwon (25 farms), Chungbuk (28 farms), Chungnam (25 farms), Gyeonbuk (26 farms), Gyeongnam (24 farms), Jeonbuk (24 farms) and Jeonnam (25 farms) and Jeju (23 farms). Also, for Korean native goat, the number of farms sampled by province was Chungbuk (21 farms), Chungnam (25 farms), Jeonbuk (23 farms) and Jeonnam (38 farms).

During sampling, medical histories were taken and the presence of respiratory signs were checked, but most of the medical histories and respiratory signs were not observed, and even the cows and goats that were positive for real-time qPCR did not have respiratory signs, so it was determined that there was no correlation.

Reviewer 2 Report

Comments and Suggestions for Authors

Based on current and hegemonic scientific information, the authors discuss the importance of analyzing the circulation of the SARS-CoV-2, through an epidemiological survey of the detection of antigens, antibodies, and virus-neutralizing antibodies in native cattle and goats from Korea. The article demonstrates to be well structured, with an appropriate methodology and a well-conducted analysis. I presented some suggestions for modifications to your text, in an attached file.

Author Response

Response to reviewers’ comments

We are pleased to resubmit a revised manuscript (no. animals-2692205) entitled “Evidence of exposure to SARS-CoV-2 in Korean native cattle and Korean native black goats in Korea” is changed to “Evidence of exposure to SARS-CoV-2 in the native cattle and black goats in Korea” for reconsideration in Journal of Animals published by MDPI as an original manuscript. We have carefully evaluated the reviewer’s comments and have provided a point-by-point response below. Changes in the manuscript have been identified by colored font. We hope that the revised manuscript meets the reviewers’ expectations at Journal of Animals.

Reviewer #2.

Comments and Suggestions for Authors

Based on current and hegemonic scientific information, the authors discuss the importance of analyzing the circulation of the SARS-CoV-2, through an epidemiological survey of the detection of antigens, antibodies, and virus-neutralizing antibodies in native cattle and goats from Korea. The article demonstrates to be well structured, with an appropriate methodology and a well-conducted analysis. I presented some suggestions for modifications to your text, in an attached file.

We thankfully and thoroughly reviewed what the reviewer’s comment on the PDF version of our paper. The reviewer’s meticulous comments were all answered and revised according to the reviewer’s comment on the main text in a Word file. Thank you.
